# Survival Outcome After Percutaneous Hepatic Perfusion with High-Dose Melphalan for Liver-Dominant Metastatic Uveal Melanoma: A 10-Year Single-Center Experience

**DOI:** 10.3390/cancers17233834

**Published:** 2025-11-29

**Authors:** Carolin M. Reiner, Martin A. Schneider, Hauke Weilert, Klara Welcker, Jochen Hoedtke, Andreas H. Mahnken, Axel Stang, Roland Brüning

**Affiliations:** 1Asklepios Campus Hamburg, Semmelweis University, 20099 Hamburg, Germany; 2Department of Diagnostic and Interventional Radiology, Asklepios Hospital Barmbek, 22307 Hamburg, Germany; mar.schneider@asklepios.com (M.A.S.); k.welcker@asklepios.com (K.W.); r.bruening@asklepios.com (R.B.); 3Department of Hematology, Oncology and Palliative Care Medicine, Asklepios Hospital Barmbek, 22307 Hamburg, Germany; h.weilert@asklepios.com; 4Asklepios Tumorzentrum, 22307 Hamburg, Germany; 5Department of Anesthesiology and Intensive Care Medicine, Asklepios Hospital Barmbek, 22307 Hamburg, Germany; j.hoedtke@asklepios.com; 6Department of Diagnostic and Interventional Radiology and Nuclear Medicine, St. Josef-Hospital, University Hospital, Ruhr University Bochum, 44791 Bochum, Germany

**Keywords:** uveal melanoma, liver metastasis, percutaneous hepatic perfusion, melphalan, chemosaturation, survival outcome

## Abstract

Melphalan-based percutaneous hepatic perfusion (M-PHP) was approved in 2023 as liver-directed treatment for patients with liver-dominant metastatic uveal melanoma (mUM), which has an overall survival (OS) ≤ 12 months. However, the reported M-PHP-related OS benefit varies between 9.6 and 27.4 months and remains poorly defined. Here, we retrospectively assessed the OS outcome in our 10-year experience comprising 99 M-PHP procedures in 38 patients with liver-dominant mUM. The analysis demonstrated a median OS of 29.1 months after first M-PHP treatment; the OS at 1, 2, and 3 years was 79.5%, 53.2%, and 28.5%, respectively. In conclusion, our center’s experience demonstrates nearly 2.5-year median OS for M-PHP-treated patients with liver-dominant mUM, which surpasses previously reported OS data. This finding implicates a possible OS benefit from M-PHP treatment in specialized centers, supports the need to account for institutional volume in clinical trials, and provides an OS reference for various novel management strategies for mUM.

## 1. Introduction

Uveal melanoma (UM) represents 3% of all melanomas and differs profoundly from cutaneous melanoma (CM) in molecular, immunologic, and clinical features [1,2,3,4,5]. Almost 50% of uveal melanoma (UM) patients develop metastatic UM (mUM), usually involving the liver as the prognostically crucial site of disease [6,7]. For those selected patients with resectable mUM, surgery provides a median overall (OS) of ~27 months [8,9,10]. However, the majority of mUM patients present with unresectable liver metastases, which is usually fatal within 3–12 months due to progressive liver metastases with hepatic failure [11,12]. For mUM patients, hepatic disease control and even slower hepatic progression matters, and therefore OS, rather than response, is the most meaningful endpoint to estimate treatment efficacy in mUM. Until recently, the median OS of mUM patients had been only ~12 months, as consistently assessed in two meta-analyses across all non-surgical treatment modalities [13,14].

While chemotherapy and immune checkpoint inhibitor (ICI) monotherapy have limited efficacy in mUM, dual-ICI regimens with anti-PD-1 and anti-CTLA-4 drugs increase median OS to ~16 months [15,16,17]. However, considering the risk of ~30–35% serious immune-related adverse events, treatment discontinuation (13.3–28.3%) and treatment-related deaths (0.7–1.0%) associated with the dual-ICI treatment [18,19], the benefit–risk ratio and cost-effectiveness deserve further exploration in mUM [20]. In the subgroup of ~35–45% HLA-A*02:01-positive mUM patients, a phase III study of tebentafusp, a bispecific HLA-directed T-cell engager, showed significantly improved OS compared to chemotherapy or ICI monotherapy (21.6 vs. 16.9 months) [21,22,23]. However, beyond HLA restriction, the high costs of tebentafusp put considerable financial pressure on the potential beneficiaries and limit clinical access [24].

Given the hepatotropism and limitations of systemic treatments, patients with liver- dominant and liver-only mUM frequently undergo liver-directed therapies (LDT), which have been associated with ~14–16-month median OS across all LDT approaches [13,25,26]. In limited disease, liver lesions can be treated with thermal ablation [27,28] or transarterial chemoembolization (TACE) [29], whereas in cases of multiple liver lesions, selective interstitial radiation therapy (SIRT) [30] and percutaneous hepatic perfusion delivery of high-dose melphalan (M-PHP) [31] are typically employed. M-PHP isolates the hepatic arterial blood flow from the systemic circulation via a double balloon vena cava catheter, saturates the liver with high-dose melphalan, and purifies the venous outflow from residual melphalan by an extracorporeal filtration system [32]. Several phase II trials support the feasibility, safety and efficacy of M-PHP in controlling liver-dominant disease in mUM, but there are inconsistencies in replicating the same median OS outcome. The reported median OS spans a broad range from 9.6 to 27.4 months [31,33,34,35,36,37,38,39,40,41], and potential relationships between the number of M-PHP cycles and OS have thus far been poorly investigated.

In the present study, we retrospectively assessed the OS outcome from our 10-year single-center experience using M-PHP as the standard of care (SOC) for patients with liver-dominant mUM. We aimed to correlate our OS estimates to OS efficacy estimates reported in the previous literature to provide information on the potential magnitude of OS benefit from M-PHP in mUM patients for clinical practice and future research.

## 2. Materials and Methods

### 2.1. Study Design

This retrospective study included consecutive adult patients (age > 18 years) with histologically confirmed liver-predominant mUM who underwent M-PHP in the radiology department of the Asklepios Hospital Barmbek between April 2014 and March 2024. Patients were allocated for M-PHP treatment based on multidisciplinary tumor board decision; the decision-making for M-PHP treatment was supported by a case-by-case discussion among surgical, radiological, interventional and oncological specialists at our institution. Inclusion criteria and specific thresholds for eligibility of M-PHP treatment were (1) Eastern Cooperative Oncology Group (ECOG) performance status of 0–1, (2) hemoglobin levels ≥ 10 g/dL, platelet counts ≥ 150,000/μL, and bilirubin levels ≤ 2 times the upper limit of normal (ULN), and (3) <70% liver tumor involvement. Patients could have limited extrahepatic disease (EHD). Any number and type of prior treatments were permitted. Patients were excluded if they had (1) extrahepatic metastases larger than 10 mm in lymph nodes (size > 10 mm measured in the short axis) and/or uncontrollable, progressive, and/or predominant EHD at other organ sites, (2) a recent history of transient ischemic attacks or heart failure with a left ventricular ejection fraction < 40%, (3) significant chronic obstructive or restrictive pulmonary disorders, and/or (4) contraindications to general anesthesia.

Baseline imaging was obtained within 6 weeks prior to M-PHP treatment with contrast-enhanced computed tomography (CT) of the chest, abdomen, and pelvis and contrast-enhanced liver magnetic resonance imaging (MRI) to determine vascular anatomy, baseline volume, and distribution of measurable disease; identify extrahepatic disease; and facilitate procedural planning. Patient histories including additional therapies, imaging findings, laboratory tests and values, clinical reports during in-hospital stay and follow-up exams were retrospectively assessed using the institutional medical records, PACS database, and tumor board protocols on e-health platforms. Extracted data were anonymized and transferred into a data sheet for statistical evaluation. The observational follow-up period ended in September 2024, with a recorded date of death or last recorded contact.

### 2.2. Procedural Details

As described in previous research [42], this approach involved isolating the hepatic circulation using a specialized filtration device, CHEMOSAT^®^ (CHEMOSAT^®^ Second Generation; Delcath Systems Inc., New York, NY, USA), to enable extracorporeal drug removal from the blood. For arterial access, a catheter was inserted into the femoral artery and advanced to the left/right hepatic artery, where a microcatheter administered melphalan at a dose of 3.0 mg/kg ideal body weight, up to a maximum of 220 mg. Venous isolation was achieved using a double-balloon catheter inserted through the contralateral femoral vein, with one balloon positioned below the hepatic veins and the other in the atrium adjacent to the diaphragm portion of the inferior vena cava. The infusion delivered the melphalan solution at a controlled rate, monitored by angiograms to ensure proper flow. This was followed by a 30 min filtration period whereby blood exiting the liver was filtered extracorporeally to remove the melphalan agent. The filtered blood was then re-infused via the right jugular vein. The procedure, performed under general anesthesia, required close postoperative monitoring, including coagulation status, blood pressure, and laboratory parameters. All patients also received G-CSF to support hematologic recovery and were monitored in intensive care for one night as a precaution.

### 2.3. Endpoints and Assessments

The primary efficacy endpoint was the median OS after M-PHP treatment. This time-to-event endpoint was defined as the time from the first PHP treatment to date of death for any case, the study cut-off date if still alive in follow-up (30 September 2024), or the last known follow-up date alive; for the latter patients, the OS was assigned censored in the analysis. The secondary exploratory endpoint analysis addressed the impact of the numbers of M-PHP cycles administered on the efficacy endpoint median OS and the risk reduction of patient death by each additional M-PHP cycle administered. For correlation to the efficacy endpoint median OS, patients were stratified in terms of two patient groups who received ≤2 or ≥3 cycles of M-PHP treatment. Adverse events (AEs) and severe AEs (SAEs) were assessed and graded according to the Clavien–Dindo (CD) [43,44], and Cardiovascular and Interventional Radiological Society of Europe (CIRSE) classification [45].

### 2.4. Statistical Analysis

Descriptive summary statistics for continuous variables were presented as median and range (minimum/maximum), and categorical variables as frequency, counts and percentages. Time-to-event variables and endpoints were assessed using Kaplan–Meier methods. OS curves and median OS, as well as corresponding 95% confidence intervals (CIs), were generated and estimated with the start date of first M-PHP application. To adjust OS probabilities to each time point, OS data were censored if patients were stated as alive at their last follow-up before the study cut-off follow-up date. The log-rank test was applied to test for significant differences between OS estimates and OS curves; two-sided *p* values of < 0.05 were considered to indicate statistical significance. Hazard ratios (HRs) from Cox proportional hazard models adjusted to the number M-PHP cycles administered were correlated with efficacy endpoints in terms of risk of death; an HR of <1 indicated a decreased risk of death. Analyses were performed with R 2025 https://r-project.org (accessed on 30 October 2025) and Rstudio IDE 2025 https://posit.co/ (accessed on 30 October 2025). [46,47]

## 3. Results

### 3.1. Patient and Study Characteristics

The study included a total of 38 patients (19 [50%] women, median age: 57.5 years; age range: 29–77 years) who underwent a total of 99 M-PHP cycles, with an average of 2.61 M-PHP cycles administered per patient (range 1–6). Exactly half of the cohort (19 patients) received 1 or 2 M-PHP cycles, while the other half (19 patients) underwent ≥ 3 cycles (range: 3–6 cycles). At first M-PHP, 34 patients (89.4%) had a liver tumor involvement of ≤25%, 6 patients (15.8%) exhibited EHD, and 20 (2.6%) had an elevated (>ULN) LDH level. Before and/or after M-PHP, 21 patients (55.2%) had received additional systemic treatments (chemotherapy, immune checkpoint inhibitors, tebentafusp), and 8 patients (21.0%) had received local liver-directed interventions (transarterial, ablative, or surgical approaches). The demographics and baseline disease characteristics of the patients are described in Table 1.

### 3.2. Survival Outcome

Patient status evaluation at the last study cut-off follow-up date on 30 September 2024 identified 23 (60.5%) patients who had died during the follow-up period. None of the deaths were considered related to M-PHP treatment, device, or procedure. Furthermore, 11 (29%) alive patients were in active follow-up, and 4 patients (10.5%) who were lost to the last follow-up date were censored as alive. These four patients are international patients who returned to their home countries after completing M-PHP treatment; all four patients were alive at their last known follow-up dates within 6 months after their last M-PHP cycle, and there was no evidence of death. After a median follow-up from the date of first M-PHP treatment of 25.8 months, the estimated median OS for the entire cohort was 29.1 months (95% CI: 18.4–38.9 months), and 1-, 2-, and 3-year OS rates were 79.5% (95% CI: 67.0–94.3%), 53.2% (95% CI: 38.2–74.0%), and 28.5% (95% CI: 15.4–52.8%) (Figure 1).

### 3.3. Number of M-PHP Cycles and Survival Outcome

Patients receiving ≥3 M-PHP cycles exhibited a median OS of 29.8 months (95% CI: 21.9–N/A), compared to 21.4 months (95% CI: 8.87–N/A) for those receiving ≤2 cycles of M-PHP; the corresponding 1-, 2-, and 3-year OS rates were 89.5% (95% CI: 76.7–100%), 59.7% (95% CI: 40.3–88.5%), and 28% (95% CI: 11.7–67.1%) versus 66.5% (95% CI: 46.2–95.7%), 44.3% (95% CI: 24.6–79.8%), and 27.7% (95% CI: 11.4–67.5%) (Figure 2). In comparative analysis, the trend towards numerically improved median OS by 8.4 months failed to show statistical significance (*p* = 0.058). Further, especially for the longer-term 3-year OS rates, the 95% CIs were largely overlapping (Figure 2). Cox proportional hazards models showed that each additional M-PHP cycle was associated with about 40% reduction in the risk of death (HR = 0.414), which supports the observed trend towards an improved OS with increasing numbers of M-PHPs administered.

### 3.4. Safety

Out of the 38 M-PHP-treated patients, 4 patients (10.5%) experienced procedure-related AEs graded ≥ 2 according to CD and CIRSE classification. Post-M-PHP AEs included one heparin-induced thrombocytopenia type II managed appropriately with alternative anticoagulation strategies (CD grade 2, CIRSE grade 2), one pulmonary embolism treated successfully by thrombolytic therapy (CD grade 2, CIRSE grade 3), one non-ST Elevation Myocardial Infarction (NSTEMI) due to an intravasal coronary thrombus treated successfully by coronary stenting and thrombolysis (CD grade 3, CIRSE grade 3), and one cerebral artery basilar embolism in conjunction with bilateral pulmonary embolism requiring interventional neuroradiological thrombectomy and systemic thrombolysis, and leaving the patient with long-term mild sequelea (CD grade 4, CIRSE grade 4). There were no procedure-related deaths. Detailed AE data are summarized in Table 2.

## 4. Discussion

In the present study, we found a median OS of 29.1 months after first administration of M-PHP in 38 patients with liver-dominant mUM. The estimated percentages of patients’ 1-, 2-, and 3-year OS were 79.5%, 53.2%, and 28.5%, respectively. The clinical significance of our study lies in providing long-term outcome data for M-PHP in mUM, supporting the procedure’s potential OS benefit. Notably, no treatment-related mortality occurred, and serious procedure-related complications were infrequent (4 out of 38 patients) and manageable, supporting the procedure’s safety. However, the OS estimates presented are observational and, as such, are exploratory and intended to provide information on potential M-PHP-related OS benefits for clinical practice and future research.

When estimating potential M-PHP-related OS benefits from our OS data, it seems helpful to refer to OS benchmarks in the evolving therapeutic landscape for mUM. First, until recently, the valid historical median OS of mUM patients was ~12 months, which has been consistently assessed in two meta-analyses across all non-surgical treatment groups [13,14]. Second, dual-ICI, which has become the first-line therapy for a considerable proportion of patients with mUM, was associated with ~16-month median OS in a recent meta-analysis [17]. Third, the median OS of 21.6 months, and the OS rates at 1, 2, and 3 years of 72%, 45%, and 27%, respectively, for tebentafusp in the landmark approval phase III trial are even slightly lower than those receiving M-PHP in our study [21,22]. Notably, use of tebentafusp is limited to ~35–45% HLA-A*02:01-positive mUM patients, and puts considerable financial pressure (additional 0.47 quality-adjusted life years [QALYs], incremental costs of USD 444.280, and incremental cost-effectiveness ratio of USD 953.230/QUALY) on the OS beneficiaries [24]. Collectively, the median OS found in our observational study (29.1 months) is more than twice as high as the historical OS (~12 months), and nearly 8 months higher than the median OS observed with the currently best pharmacotherapy of mUM (21.6 months). Although cross-trial efficacy comparison is methodically difficult, these OS differences are large enough to be considered clinically meaningful, and they apply to ~90% of mUM patients, as the liver is the primary (~90%), only (~50%) and prognostically crucial site of disease [6,7,11,12]. Notably, M-PHP received approval in 2023 for liver-dominant mUM based on the FOCUS trial, which demonstrated consistent efficacy across patients with exclusively hepatic and limited extrahepatic disease [31].

The multicenter phase III FOCUS (M-PHP) and SCANDIUM trial (M-IHP) evaluated hepatic perfusion with high-dose melphalan in mUM patients compared to a control group receiving best alternative care (BAC) according to investigators’ choice (TACE, chemotherapy, ICI monotherapy) [48,49]. Both trials found a clinically meaningful albeit not significantly improved median OS by ~4 months in favor of M-PHP (18.5 vs. 14.5 months) or M-IHP (21.7 vs. 17.6 months). The final results from these two phase III trials, published in 2025, provide the current strongest evidence for a disease-modifying effect and OS gain from hepatic perfusion with high-dose melphalan in mUM patients. Notably, the FOCUS trial closed early due to slow recruitment (85 patients enrolled vs. 240 planned), which illustrates the difficulty to conduct randomized trials aimed at comparing treatments head-to-head in low-prevalence mUM.

There were no M-PHP procedure-related deaths in our study cohort. However, post-M-PHP thromboembolic events were a notable complication in 10.5% of our patients, indicating cautious use of protamine sulfate infusion to reverse heparinization in the post-M-PHP phase. A recent meta-analysis found, on average, a similar OS (17.1 vs. 17.3 month) but higher 30-day-mortality (5.5% vs. 1.8%) for surgical M-IHP compared to minimally invasive M-PHP as loco-regional treatment for liver metastases in mUM [50]. In addition to that, M-PHP has advantages over I-PHP in terms of patient recovery and procedure repeatability. A recent systematic review reported a median OS of 12.3 months for SIRT, an alternative LDT for unresectable liver metastases in mUM [51]. Further, a retrospective single-center study found a significantly longer OS for mUM patients treated with M-PHP than with SIRT (17.2 vs. 10.0 months) [33]. Altogether, these data support M-PHP as the preferred whole-liver LDT to target evident and occult liver metastases in mUM, either for safety (vs. M-IHP) or efficacy (vs. SIRT) reasons.

Over the last decade, published OS data on M-PHP for mUM have been heterogeneous. An earlier phase III trial showed no difference between M-PHP and BAC in mUM patients (10.6 vs. 10.0 months) [34], and non-randomized studies investigating the feasibility, safety, and efficacy of M-PHP in mUM reported different median OS results, ranging between 9.6 and 27.4 months [31,33,35,36,37,38,39,40,41]. The inconsistencies in replicating the same OS outcome may come from multiple sources, such as small sample size, differences in study design and/or conduct, variability of patient selection, study cohorts and/or prognostic factors, and technological improvements of the M-PHP procedure over time [26,32,39]. The same between-study variations bias individual cross-trial comparisons, but it is legitimate to compare our median OS (29.1 months) to the aggregated median OS (17.3 months) found in a meta-analysis of M-PHP trials in mUM [50]. Compared to this fairly objective effect size for M-PHP, and the OS estimate in the approval-relevant FOCUS trial [48], the median OS estimate in our study is even longer. This OS difference may in part be attributed to center experience, case volume, and multidisciplinary patient selection. The impact of such factors on the OS outcome of the demanding M-PHP procedure-related patient care may be underestimated and should receive greater attention in future research.

In this study, a total of 38 patients received 99 M-PHP procedures. While our average number of M-PHP cycles per patient (2.6) is within the range (1.8–4.0) of previous studies [31,33,35,36,37,38,39,40,41], we found that each additional M-PHP cycle was associated with ~60% reduced risk of death (HR = 0.414). Moreover, the median OS was numerically 8.4 months higher in patients treated with 3–6 (29.8 months) versus 1–2 M-PHPs (21.4 months). This OS difference showed a trend, but failed to achieve statistical significance (*p* = 0.058), which could in part be attributed to the limited sample size. Otherwise, the almost similar OS rates and overlapping 95% CIs at 3 years (28.0% [11.7–67.1%] vs. 27.7% [11.4–67.5%]) demand caution as to whether the OS trend will persist in the long term. The literature provides little data on the correlation between the number of M-PHPs and OS. Further, there are likely confounding effects from patient management at play, making it difficult to disentangle the M-PHP cycle number as a cause or effect of prolonged OS. For example, the landmark FOCUS trial attempted to complete six M-PHP cycles per protocol (45% of patients completed six cycles) and discontinued M-PHP treatment in case of disease progression [31,48]. In contrast, we personalized the treatment goals and indications before each M-PHP cycle in a multidisciplinary manner, which could also mean continuing M-PHP to slow hepatic disease progression. However, association does not mean causation, and our data do not prove that additional M-PHP cycles cause longer survival. Reverse causation could also be possible, and that slower-progressing patients can receive more M-PHP cycles. However, M-PHP holds an advantage in its repeatability, and our observed trend towards improved OS with higher numbers of M-PHP treatments warrants further investigation in larger, prospective trials.

Our study has limitations. First, major limitations stem from its retrospective design and lack of a control group. Thus, the OS efficacy analyses presented in this study are exploratory. Second, the single-center nature and our strict selection criteria introduce potential selection bias. This could limit the generalizability of our results to a broader mUM population. Third, the sample size is modest, and ~55% of patients received additional therapies before or after M-PHP. This could confound the observed OS outcomes; however, the high diversity of additional periprocedural treatments, along with the small sample and subgroup sizes, limited to produce meaningful data on the effect of additional therapies. Altogether, all efficacy analyses presented in this report are exploratory. Further prospective, larger-scale, multicenter studies in independent patient populations are required to explore the role of additional therapies, to evaluate whether more M-PHP cycles cause longer OS, and to verify the extent to which the findings presented here are generalizable. However, as for its strengths, the OS calculations in our analysis are based on sufficient long-term follow-up, and the magnitude of M-PHP-associated OS effects contributed to the significance of findings. Despite the aforementioned study limitations, we believe that our OS data may provide a source for cross-trial comparison to reported OS efficacy estimates and/or OS advances of various novel management strategies for mUM. In particular, it might be valuable to compare our OS data (median OS: 29.1 months; 2-years OS: 53.2%) with OS data from recent studies on combining M-PHP with immunotherapies and/or targeted therapies [52,53,54], immunotherapy combinations in tebentafusp-naive and tebentafusp-pre-treated mUM cohorts [55], and novel strategies such as (PRAME)-directed T-cell therapy in HLA-A*02+ mUM patients [56].

## 5. Conclusions

In the challenging treatment of liver-dominant mUM, we found a median OS of 29.1 months after the first M-PHP administration. This OS result is more than twice as high as the historical OS that was valid until 2020 (~12 months), and reaches the OS data for tebentafusp in the approval landmark phase III trial in 2021. Limitations of our analysis include its exploratory character, retrospective design, and the lack of a control group. A prospective evaluation is necessary to confirm our findings. Nevertheless, our long-term outcome results suggest a clinically meaningful OS benefit from M-PHP in the treatment of patients with liver-dominant mUM. Further, our results may also indicate that institutional experience, operators’ skills, case volume, and multidisciplinary patient selection will be essential to fully harness the OS benefits of M-PHP in clinical practice and future research.

## Figures and Tables

**Figure 1 cancers-17-03834-f001:**
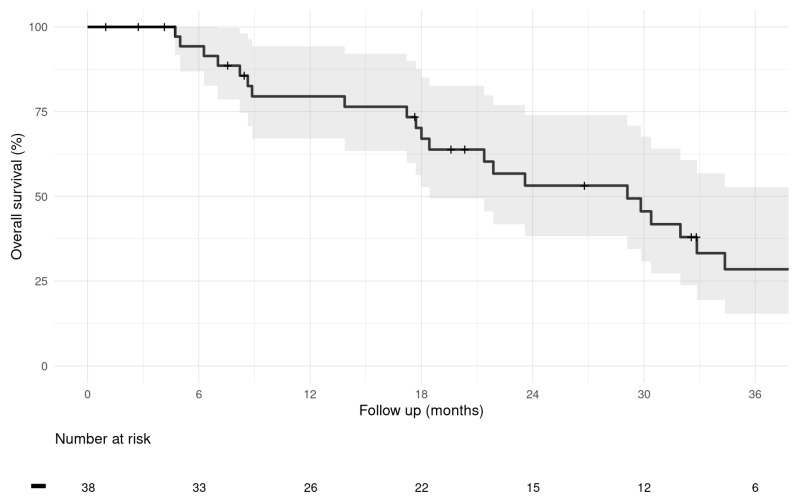
Kaplan–Meier graph of overall survival of the entire study population calculated from first M-PHP treatment. (Median; 95% CI).

**Figure 2 cancers-17-03834-f002:**
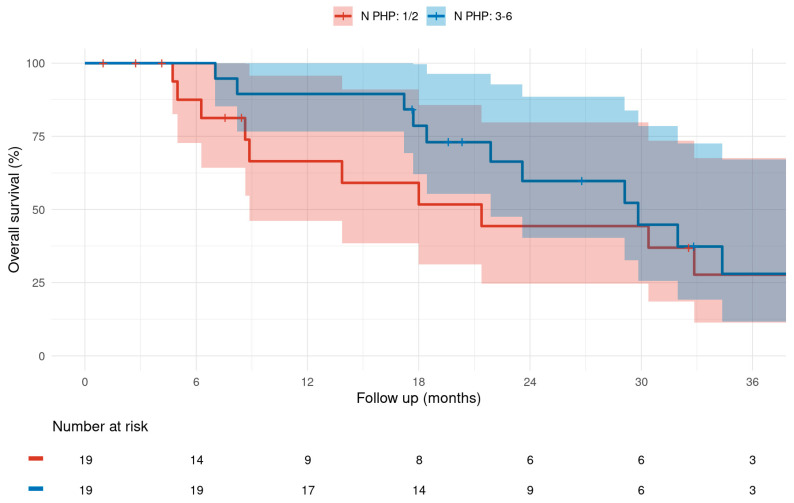
Kaplan–Meier graph of overall survival difference calculated from first M-PHP treatment for a total of 1–2 versus 3–6 M-PHP cycles administered. (Median; 95% CI).

**Table 1 cancers-17-03834-t001:** Baseline patient characteristics of patients undergoing melphalan-based percutaneous hepatic perfusion (M-PHP).

Parameters	Entire Cohort	
Sample size (n)		38
Age, ^1^ year		57.5 (29–77)
Male ^3^	Count (%)	19 (50)
ECOG-PS, ^3^ score ≤1	Count (%)	38 (100)
Extent of liver involvement at first M-PHP		
≤10% ^3^	Count (%)	29 (76.3)
10–25% ^3^	Count (%)	5 (13.2)
25–50%, ^3^	Count (%)	3 (7.9)
>50% ^3^	Count (%)	1 (2.6)
Extrahepatic disease at first M-PHP		
Yes ^3,4^	Count (%)	6 (15.8)
Median time since primary diagnosis to first M-PHP		
Time, ^2^ years (initial diagnosis uveal melanoma—last contact)	Median (IQR)	3.98
Time, ^2^ years (initial diagnosis hepatic metastasis/es—last contact)	Median (IQR)	1.95
Lactate dehydrogenase (LDH) before first M-PHP		
>ULN ^5^ at first M-PHP ^3^	Count (%)	20 (52.6)
Additional systemic therapy before/after M-PHP		
Chemotherapy, ^3,6^	Count (%)	7 (18.4)
Immune checkpoint inhibitors, ^3,7^	Count (%)	11 (28.9)
Tebentafusp only ^3^	Count (%)	3 (7.9)
Additional liver-directed therapy before/after M-PHP		
Transarterial approaches, ^3,8^	Count (%)	2 (5.26)
Ablative approaches, ^3,9^	Count (%)	2 (5.26)
Surgical resection ^3^	Count (%)	4 (10.5)
No additional treatments		
Best supportive care (BSC) ^3^	Count (%)	17 (44.7)

^1^ Mean (standard deviation) [normal data distribution]; ^2^ median (quartile 1, quartile 3) [outside normal distribution]; ^3^ count (percentage); ^4^ lymph node, lung, bone, other visceral, soft tissue, brain; ^5^ ULN, upper limit of normal; ^6^ dacarbazine, temozolomide, gemcitabine, platinum, and/or taxane; ^7^ dual-agent ICI ipilimumab and nivolumab, or monotherapy ipilimumab, nivolumab, or pembrolizumab; ^8^ transarterial chemoembolization (TACE), or selective internal radiotherapy (SIRT); ^9^ radiofreqency ablation (RFA), microwave ablation (MWA).

**Table 2 cancers-17-03834-t002:** Adverse events (AEs) graded ≥ 2 according to the Clavien–Dindo (CD) and Cardiovascular and Interventional Radiological Society of Europe (CIRSE) classification after melphalan-based percutaneous hepatic perfusion (M-PHP) for liver-dominant metastatic uveal melanoma.

Grade	CD	No.	Grade	CIRSE	No
1	Deviation from normal postoperative course without need for pharmacological treatment or surgical, endoscopic and radiological interventions	NA	1	Complication during the procedure which could be solved within the same session; no additional therapy, no post-procedure sequelae	NA
2	Requiring pharmacological treatment with drugs other than such allowed for grade I	2 ^a,b^	2	Prolonged observation including overnight stay as a deviation from the normal post-therapeutic course <48	1 ^a^
3	Requiring surgical, endoscopic or radiological intervention	1 ^c^	3	Additional post-procedure therapy or prolonged hospital stay (48 h) required; no post-procedure sequelae	2 ^b,c^
4	Life-threatening complication (including central nerve system complications) requiring Intensive Unit Care	1 ^d^	4	Complication causing a permanent mild sequelae (resuming work and independent living)	1 ^d^
5	Death	0	5	Complication causing permanent severe sequelae (requiring ongoing assistance in daily life)	0
			6	Death	0
All	Grades 2–5	4		Grades 2–6	4

^a^ Heparin-induced Thrombocytopenia Type II managed appropriately with alternative anticoagulation strategies (CD grade 2, CIRSE grade 2). ^b^ Pulmonary embolism treated successfully by thrombolytic therapy (CD grade 2, CIRSE grade 3). ^c^ Non-ST Elevation Myocardial Infarction due to intravasal coronary thrombus treated successfully by coronary stenting and thrombolysis (CD grade 3, CIRSE grade 3). ^d^ Cerebral artery basilar embolism in conjunction with bilateral pulmonary embolism requiring interventional neuroradiological thrombectomy and systemic thrombolysis, and leaving the patient with long-term mild sequelae (CD grade 4, CIRSE grade 4).

## Data Availability

The data presented in this study are available on reasonable request from the corresponding author. The data are not publicly available due to privacy.

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
