# Peer review of "Survival Outcome After Percutaneous Hepatic Perfusion with High-Dose Melphalan for Liver-Dominant Metastatic Uveal Melanoma: A 10-Year Single-Center Experience"

_cancers, 2025, doi:10.3390/cancers17233834_

Round 1

Reviewer 1 Report

Comments and Suggestions for Authors

General: This manuscript reports a retrospective single-center analysis of M-PHP in 38 patients with liver-dominant mUM) for 10-years. The topic is important in interventional oncology, given the historically poor prognosis of mUM and strengthens recent interest in liver-directed therapies. A median overall survival (OS) of about 29.1 months after first M-PHP treatment is reported, which is substantially longer than traditional outcomes. Higher OS in patients receiving ≥3 M-PHP cycles, causing ~60% decrease in hazard of death (HR ~0.4) per cycle. The data are clearly presented with Kaplan-Meier survival curves and tables of baseline characteristics and adverse events.

Overall, the significance of this work lies in providing long-term outcome data for M-PHP in mUM, supporting the procedure’s potential survival benefit. The manuscript’s strengths include the extensive follow-up, comprehensive context provided (referencing recent trials like FOCUS and SCANDIUM), and the systematic reporting of safety using standardized classifications. Of note, no treatment-related mortality occurred, and serious procedure-related complications were infrequent (4 out of 38 patients). The writing quality is generally good, and required structure is correct. The data presentation is mostly clear, with well-organized tables and figures.

The weaknesses stem from its retrospective design and lack of a control group. The single-center nature and strict selection criteria introduce potential selection bias. The sample size (38 patients) is modest, and many patients received additional therapies before or after M-PHP, which could confound the survival outcomes. While the authors acknowledge several of these limitations in the discussion, some results and comparisons (e.g. improved OS with more M-PHP cycles) warrant cautious interpretation. 

Specific: 

  1. Please add a clear statement that no direct concurrent control was present. This will temper the conclusion and ensure readers understand the evidence level of the study.
  2. Patient Selection and Generalizability: highly selected population, Please discuss the generalizability of the results to broader mUM populations.
  3. Additional Therapies (Confounding factors): clarify when these additional therapies were given (before, interspersed with, or after M-PHP) and discuss their potential impact.
  4. Number of M-PHP Cycles – Causation vs. Association: The finding that ≥3 M-PHP cycles correlates with longer median OS (29.8 vs 21.4 months) is interesting, but it may reflect patient selection (patients who survive longer can receive more cycles, whereas early progressors stop at 1–2 cycles). I think a more cautious interpretation is warranted. Data does not prove that additional cycles cause longer survival. Reverse causation could be possible, that slower progressing patient can receive more cycles.
  5. Efficacy Endpoints – Response and Progression: tumor response rates or progression-free survival (PFS) should be reported (e.g., RECIST tumor response, or hepatic PFS).
  6. Procedural Details: specify the melphalan dosing regimen used for M-PHP. The dose of “high-dose melphalan” is not stated.
  7. Describe hematologic toxicity more detailed. What were the rates of Grade 3–4 cytopenias, was transfusion required?
  8. Data Presentation – Figures and Tables:  Figure 1 and 2 seem to have low resolution, please check. Table 2 is complex, please simplify or explain more detailed in captions.
  9. Language and Wording: "became approved in 2023" better might be "was approved". Please scan the text for minor typos (e.g., “the la‐er” instead of “the latter” on page 5). “M-PHP” or “PHP” more consistently. 
  10. Discussion – Scope and Emphasis: streamline any repetitive points. Phrases like “more than twice as high as the historical OS” are factually correct but could be followed by a reminder that such comparisons are observational.

Author Response

General: This manuscript reports a retrospective single-center analysis of M-PHP in 38 patients with liver-dominant mUM) for 10-years. The topic is important in interventional oncology, given the historically poor prognosis of mUM and strengthens recent interest in liver-directed therapies. A median overall survival (OS) of about 29.1 months after first M-PHP treatment is reported, which is substantially longer than traditional outcomes. Higher OS in patients receiving ≥3 M-PHP cycles, causing ~60% decrease in hazard of death (HR ~0.4) per cycle. The data are clearly presented with Kaplan-Meier survival curves and tables of baseline characteristics and adverse events.

Overall, the significance of this work lies in providing long-term outcome data for M-PHP in mUM, supporting the procedure’s potential survival benefit. The manuscript’s strengths include the extensive follow-up, comprehensive context provided (referencing recent trials like FOCUS and SCANDIUM), and the systematic reporting of safety using standardized classifications. Of note, no treatment-related mortality occurred, and serious procedure-related complications were infrequent (4 out of 38 patients). The writing quality is generally good, and required structure is correct. The data presentation is mostly clear, with well-organized tables and figures.

The weaknesses stem from its retrospective design and lack of a control group. The single-center nature and strict selection criteria introduce potential selection bias. The sample size (38 patients) is modest, and many patients received additional therapies before or after M-PHP, which could confound the survival outcomes. While the authors acknowledge several of these limitations in the discussion, some results and comparisons (e.g. improved OS with more M-PHP cycles) warrant cautious interpretation. 

Authors response: We thank the reviewer for this overall positive impression, critical comments, and the helpful suggestions.   

Specific: 

  1. Please add a clear statement that no direct concurrent control was present. This will temper the conclusion and ensure readers understand the evidence level of the study.

Authors response: We agree that a more clear startement on the lacking control group enhances the readers understanding on the evidence level of the study. We include this statement in the first and last paragraph in the discusion section. Further, we discuss the limitations of our work in more detail under the paragraph limitations in the discussion section. Specifically, we point out the exploratory character of our analysis and the need to confirm our results in prospective trials (for details, please see changes in the discussion section).

  1. Patient Selection and Generalizability: highly selected population, Please discuss the generalizability of the results to broader mUM populations.

Authors response: We agree that the generalizability of our exploratory data requires a more extended discussion. We address this point in the now more detailed discussion of the study limitations.

  1. Additional Therapies (Confounding factors): clarify when these additional therapies were given (before, interspersed with, or after M-PHP) and discuss their potential impact.

Authors response: We generally agree that the peri-intervential management could have a potential impact on the outcome results. However, nearly half of the patients (n = 17; 44.%) did not receive any additional treatments. Further, the additional treatments span a very broad range of very different local and systemic therapies. This heterogeneity reflects that there is no standard of care for mUM patients and limits a meaningful detailed discussion on the potential impact of additional therapies. We therefore present the cumulative data on additional therapies and refrain from further detailed differentiations. However, we adress this point more cleary in limitations (discussion section).

  1. Number of M-PHP Cycles – Causation vs. Association: The finding that ≥3 M-PHP cycles correlates with longer median OS (29.8 vs 21.4 months) is interesting, but it may reflect patient selection (patients who survive longer can receive more cycles, whereas early progressors stop at 1–2 cycles). I think a more cautious interpretation is warranted. Data does not prove that additional cycles cause longer survival. Reverse causation could be possible, that slower progressing patient can receive more cycles.

We thank the reviewer for these critical comments and helpful suggestions. We address this important objection and consideration and state them in the discussion section. For details, please see the additions in the paragraphs on the number of M-PHP cycles and study limitations in the discussion section.  

  1. Efficacy Endpoints – Response and Progression: tumor response rates or progression-free survival (PFS) should be reported (e.g., RECIST tumor response, or hepatic PFS).

Authors reponse: We understand that additional data (response rates, PFS) could complement our data. However, we have deliberately refrained from doing so, as we believe these data would not improve the core message and clinical significance of our results. In our opinion data on local tumor response and hepatic PFS is not the best parameter to qualify clinical use of M-PHP for treatment of  mUM patients. In our clinical understanding patients OS benefit matters most  as OS is generally the crucial outcome measure in clinical trials in oncology and in the field of interventional oncology, especially if the potential clinical benefit (efficay and safety) of local (loco-regional) treatments is investigated in the context of systemic cancer disease.

  1. Procedural Details: specify the melphalan dosing regimen used for M-PHP. The dose of “high-dose melphalan” is not stated.

Authors response: We state the melphalan dose used for M-PHP in the M&M section: „For arterial access, a catheter was inserted into the femoral artery and advanced to the left/right hepatic artery, where a microcatheter administered melphalan at a dose of 3.0 mg/kg ideal body weight, up to a maximum of 220 mg.“ (please see M&M section).

  1. Describe hematologic toxicity more detailed. What were the rates of Grade 3–4 cytopenias, was transfusion required?

Authors response: We deliberately refrained from reporting laboratory-related but clinically irrelevant complications. We limited ourselves to reporting complications requiring treatment (please see Table 2). There was no case with anemia and/or thrombocytopenia requiring prophylactic or therapeutic transfusion. Short comment: Our criteria for determining whether or not to administer transfusions were extremely strict. The indication for transfusion was not based solely on laboratory values, but only on laboratory values ​​in conjunction with clinical symptoms. The restrictive transfusion management was based on our long-term experience with patients under myeloablative treatment for acute leukemia. All anemia cases were asymptomatic, and all thrombocytopenia cases were without clinical relevance to bleeding or heavy bleeding signs. Therefore, we did not administer any prophaylactic or therapeutic transfusions. We believe that the deliberate omission of clinically irrelevant laboratory „side effects“ supports the clarity of our focus and core messages on clinically meaningful endpoints for efficacy (clinically most important: OS) and safety (clinically most important: severe complications requiring treatment).   

  1. Data Presentation – Figures and Tables:  Figure 1 and 2 seem to have low resolution, please check. Table 2 is complex, please simplify or explain more detailed in captions.

We checked the resolution We have checked the resolution of Figures 1 and 2 and consider it sufficient. Should the resolution actually be too low, we can try sending Figures 1 and 2 in a higher resolution. We have simplified Table 2 (percentages omitted, they are mentioned in the manuscript text).  

  1. Language and Wording: "became approved in 2023" better might be "was approved". Please scan the text for minor typos (e.g., “the la‐er” instead of “the latter” on page 5). “M-PHP” or “PHP” more consistently. 

We thank the reviever for these critical comments. We changed the mentioned phrases accordingly.

  1. Discussion – Scope and Emphasis: streamline any repetitive points. Phrases like “more than twice as high as the historical OS” are factually correct but could be followed by a reminder that such comparisons are observational.

We thank the reviever for these helpful suggestions. We now emphasize the observational nature of our data analysis more strongly at several points in the discussion section.  

Reviewer 2 Report

Comments and Suggestions for Authors

Thank you for your excellent work and summary of treatment results, including safety. One point for clarification in Section 2.1 Study Design. It is stated: Patients were excluded if they had (1) extrahepatic metastases larger than 10 mm in lymph nodes. Note, per RECIST, LNs below 10mm are considered normal. Does it mean that patients with LNs in the short axis equal 10mm were eligible? I would suggest rewriting this statement for clarity. No other comment from me. Thank you again. Congratulations

Author Response

Thank you for your excellent work and summary of treatment results, including safety. One point for clarification in Section 2.1 Study Design. It is stated: Patients were excluded if they had (1) extrahepatic metastases larger than 10 mm in lymph nodes. Note, per RECIST, LNs below 10mm are considered normal. Does it mean that patients with LNs in the short axis equal 10mm were eligible? I would suggest rewriting this statement for clarity. No other comment from me. Thank you again. Congratulations

Authors response: We thank the reviewer for this very positive feedback. We added the term „short axis“ in our M&M section.    

Reviewer 3 Report

Comments and Suggestions for Authors

This single-center study reports a 29.1-month median overall survival in 38 patients with liver-dominant metastatic uveal melanoma treated with M-PHP, far exceeding previous outcomes. The results are clearly presented and support a meaningful survival benefit. However, the retrospective design, small sample size, and potential patient selection bias limit its generalizability. The findings are significant but require validation in a multi-center setting to confirm if this outstanding efficacy is reproducible elsewhere, or could be validated using a randomized clinical trial to test its efficacy against the current standard of care.

Author Response

Authors response:

We thank the reviewer for this overall positive feedback and the helpful suggestions.   

We agree that a more clear statement on the limitations (retrospective design, small sample size, potential selection bias, lacking control group, etc.) enhances the readers understanding on the evidence level of the study. Accordingly, we added this statement in the first and last paragraph in the discusion section. Further, we discuss the limitations of our work in more detail under the paragraph limitations in the discussion section. Specifically, we point out the exploratory character of our analysis and highlight more clearly that further prospective, larger-scale studies are needed to confirm our results and the generalizability of our findings in independant mUM patient populations.

Round 2

Reviewer 1 Report

Comments and Suggestions for Authors

Thank you for the revision. Most of the key comments raised during the review have been fully and satisfactorily addressed. The manuscript has clearly improved and gained value as a result.

Reviewer 3 Report

Comments and Suggestions for Authors

I have no more questions.